# Experiences of redeployed healthcare workers in the fight against COVID-19 in China: A qualitative study

Houshen Li[1☯], Yifan Cui[2☯], Nikolaos Efstathiou[3], Bo Li[2]*, Ping Guo[1,3]*

**1** Cicely Saunders Institute of Palliative Care, Policy & Rehabilitation, Florence Nightingale Faculty of Nursing, Midwifery, and Palliative Care, King's College London, London, United Kingdom, **2** School of Nursing and Health, Henan University, Kaifeng, Henan Province, P.R. China, **3** School of Nursing and Midwifery, Institute of Clinical Sciences, College of Medical and Dental Sciences, University of Birmingham, Birmingham, United Kingdom

☯ These authors contributed equally to this work.
* p.guo@bham.ac.uk (PG); 10210022@vip.henu.edu.cn (BL)

**Data Availability Statement:** Data cannot be shared publicly because of risk of violating privacy. Data are available from the Henan University Institutional Data Access / Ethics Committee for

## Abstract

### Introduction

Public health responses were triggered while COVID-19 was spreading. China redeployed healthcare workers to serve the most vulnerable populations and communities in the initial epicentre—Wuhan. However, it is not known how redeployment processes impacted on healthcare workers in a pandemic crisis.

### Aims

To explore the experiences and needs of frontline healthcare workers who were redeployed to care for COVID-19 patients in Wuhan, China, and understand the long-term impacts of the redeployment experience on their work and life.

### Methods

A qualitative study was conducted with redeployed healthcare workers using semi-structured interviews and thematic analysis. This study is reported in accordance with the consolidated criteria for reporting qualitative research (COREQ) guidelines.

### Findings

A total of 20 redeployed healthcare workers (13 nurses and seven physicians) participated, and four themes were generated: (1) Initial feelings and emotions of redeployment—Participants experienced worries and concerns, a sense of isolation and loneliness on their arrival to the epicentre. (2) 'It is like a war zone'—Healthcare workers faced a range of risks and challenges of caring for COVID-19 patients in Wuhan in the context of resource strain. (3) Uncertainty and coping strategies in patient care—Despite the hardships experienced, participants continued to deliver high-quality patient care including psychological care and palliative care, good communication and building mutual trusting relationships. (4) Reflection

researchers who meet the criteria for access to confidential data. Data access requests may be sent to Medical Ethics Committee, College of Medicine, Henan University, the North section of Jinming Street, Kaifeng, Henan Province, China Postcode: 475004 Phone: +86 0371 22951718 Email: hdyxykyb@163.com.

**Funding:** The author(s) received no specific funding for this work.

**Competing interests:** The authors have declared that no competing interests exist.

and far-reaching impacts of caring for COVID-19 patients—Participants felt motivated and encouraged as efforts were recognised by the government and wider society.

## Conclusions

Redeployed healthcare workers shared their unique needs and experiences of coping with redeployment and challenges they faced in the context of resource strain, which has significant implications for policy and future practice. The reality of a pandemic may reduce healthcare workers' willingness to work due to various reasons including inadequate preparedness of facilities and workplace safety. It is important to support frontline healthcare workers in order to maintain an adequate healthcare workforce in pandemic crises. Continuously evolving pandemic circumstances and uncertainty highlight the importance of an organized national pandemic response plan for subsequent waves of COVID-19 and future pandemics.

## Introduction

COVID-19 is characterised by high transmission rates, triggering public health responses globally [1, 2]. Hundreds of millions of individuals were infected worldwide, resulting in deaths, a substantial indirect influence on other diseases, and posing major public health and governance challenges in terms of economic, medical, and health infrastructure [3–7]. People with chronic underlying diseases are vulnerable to the virus and have experienced worse outcomes. Severe disease can lead to heart, and respiratory failure, acute respiratory distress syndrome, or even death as well as serious psychological outcomes [8–10].

The rapid escalation in the number of COVID-19 infections requiring treatment in hospitals has resulted in insufficient healthcare resources [11]. The overwhelming impact of COVID-19 stretched health systems' capacity and brought adverse effects to healthcare providers, including the risk of infection [12–14]. There will probably be a high burden of COVID-19 in countries with fragile healthcare systems, lack of access to clean water and disinfectants, poor outbreak preparedness, severe shortages in personal protective equipment (PPE) and medical technology, challenges in enforcing physical distancing regulations, and reliance on informal employment [15]. In such countries, patients with severe COVID-19 who are unable to receive acute care in hospitals may suffer and die, being cared for by family members without PPE and access to relevant information, training, or palliative care resources [16].

At the early stage of the COVID-19 pandemic, in the face of a previously unknown virus, China rolled out the most ambitious, agile and aggressive disease containment effort in history [17, 18]. In Wuhan, Hubei province of China—the initial epicentre of COVID-19, general wards were quicky modified into isolation wards, and healthcare workers stepped up to provide care for patients with COVID-19 in 'fever clinics'. Respiratory and infection disease units were overwhelmed by an increasing number of suspected and confirmed cases.

China launched a national emergency response and initiated a range of measures to mitigate the transmission of COVID-19 and reduce the impact of the outbreak on healthcare systems, including redeployment of healthcare resources to serve the most vulnerable populations and communities [19]. During the COVID-19 crisis, staff redeployment has been extensive and varied. As of February 29, 2020, central government redeployed over 42,000 healthcare workers (including 28,600 nurses) and vital PPE supplies across China to Hubei province [20]. Staff with different specialities have been redeployed into high-risk areas from their relatively safe 'home' wards [21].

Redeployed healthcare workers did not have adequate PPE and witnessed patients and relatives in distress. They had to deal with risk and uncertainty, quicky adapted to changes at work, and experienced persistent worry and fear about themselves, their family and friends as they could contact or transmit the disease [22, 23]. In addition, working in medical facilities serving COVID-19 patients was stressful and drained some healthcare workers physically and emotionally [24, 25]. Indeed, the fear of getting infected or infecting family and friends, the hefty workload, the intermittent shortage of PPE, and the need to take increased precautions during medical examinations and in the operative fields created enormous psychological burden to healthcare workers [26, 27].

The longer-term consequences of this crisis for healthcare workers are not clear. In a cohort of healthcare workers caring for severe acute respiratory syndrome (SARS) patients, significantly higher levels of burnout, psychological distress and posttraumatic stress were reported after the outbreak ended [28]. It is well recognised that the wellbeing of staff impacts on patient care. In recent systematic reviews, staff burnout was associated with lower patient satisfaction, less professionalism, and higher levels of clinical errors [29, 30].

Healthcare workers at the front line of the COVID-19 outbreak are exposed to hazards that put them at risk of infection. As of February 11, 2020, among 3019 healthcare workers in the 422 medical facilities serving COVID-19 patients who have been infected, 1716 were diagnosed with COVID-19 in China [31]. Overall, 14.6% of confirmed cases among healthcare workers were classified as severe or critical and five deaths were recorded [31]. In addition, healthcare workers caring for COVID-19 inpatients were under extraordinary stress related to high risk of stigmatization, understaffing, and uncertainty [32]. Quantitative studies have shown that frontline healthcare workers treating patients with COVID-19 have greater risks of mental health problems, such as anxiety, depression, insomnia, and stress [33]. Transmission from healthcare workers to their family members is widely reported [34].

Understanding what healthcare workers have experienced when redeployed during the COVID-19 pandemic is important to allow the development of supporting mechanisms, which may have huge implications for patient care [21, 35, 36]. However, so far, there is limited research on redeployment processes and how these could impact on healthcare workers in a pandemic crisis where large swathes of staff needed to be mobilised quickly to work in an epicentre. Hence, in this study we aimed to explore the experiences and needs of redeployed frontline healthcare workers caring for COVID-19 patients in Wuhan, China, and understand the long-term impacts of the redeployment experience on their work and life. The specific objectives were:

1. To explore distress, issues and challenges experienced, and coping mechanisms used from the healthcare workers' perspective.

2. To explore experiences and views towards caring for COVID-19 patients at the end of their life.

3. To explore care delivery and health system changes in response to the COVID-19 pandemic, identify areas for improvement, and make recommendations for policy makers and healthcare providers to improve preparedness for future pandemics.

## Methods

### Design

This was a descriptive qualitative study which used semi-structured interviews and thematic analysis and is reported in accordance with the consolidated criteria for reporting qualitative research (COREQ) guidelines to enhance trustworthiness and methodological rigour [37].

This study was reviewed and approved by the Research Ethics Committee at the University of Birmingham (ERN_20–1134) and Henan University (HUSOM2020-252).

## Setting and participants

The study was conducted in the First Affiliated Hospital of Henan University, where the healthcare workers were redeployed from. The First Affiliated Hospital of Henan University is a comprehensive university teaching hospital integrating medical treatment, teaching, research, prevention, and health care. The healthcare workers were redeployed on January 26, 2020 (the second day after Lunar New Year) to the epicentre of the COVID-19 infections and entered the isolation ward of Gutian Branch of Wuhan Fourth Hospital which was identified as one of the designated hospitals for COVID-19 in Wuhan.

Physicians or nurses who were redeployed and had worked in the front line providing direct care and treatment for patients with COVID-19 were recruited for the study. All participants were able to communicate in Chinese and able to give informed consent. Purposive sampling was used aiming to achieve heterogeneity in age, gender, professional groups and working experience. Eligible healthcare providers were identified and had the study explained to them by the researcher (YC). They were provided with information sheets by email, and asked whether they would like to participate in the study or not. If they were interested in participating, they were approached (or contacted by telephone if face to face meetings were impossible during the current pandemic) by the researcher who provided further study details and answered any questions they had about the study. All potential participants were given at least 24h to consider if they would like to participate or not. Those who decided to participate gave written consent. Participants were informed that they could withdraw from the study at any time without giving a reason. The deadline for withdrawal of participant data was four weeks after each interview, as after that time analysis would have started and would be impossible to remove the analysed data.

## Data collection

Face-to-face, semi-structured in-depth interviews were conducted in the participants' preferred settings between September 1, 2020 and May 31, 2021. This type of data collection helped to elicit deep and rich information in relation to the aim and objectives of the study. The interviewer (YC) had no relationship with any of the participants at the time of consenting and interviewing. Each interview was anticipated to last 30–60 mins. A topic guide was developed based on reviews of evidence on experiences and needs of healthcare workers in pandemics, refined by the research team, and was used to guide the interviews (S1 File).

The topic guide addressed (1) the experiences of redeployment and providing direct care for COVID-19 patients, (2) personal priorities, main needs, and concerns, (3) distress being experienced, issues and challenges being faced, and ways of coping, (4) experiences and views towards caring for end-of-life COVID-19 patients (e.g., symptoms, multi-dimensional needs, preferences of those patients, difficult decision making towards life-sustaining treatments), (5) current care delivery and health system changes in response to COVID-19, and areas for improvement, and (6) recommendations to improve preparedness for future pandemics.

Data collection continued until data saturation was reached (i.e., no new findings were identified in line with the study aim). All interviews were conducted in Chinese and digitally audio-recorded, anonymised, and finally transcribed verbatim. The first five interviews (one-fourth of all interviews) were translated into English to allow topic guide refinement and initial development of a coding framework by the research team, who were not all native Chinese. Field notes were made during each interview.

## Data analysis

Data were jointly coded by HL and YC, and analysed using inductive thematic analysis in QSR NVivo 12 [38]. PG and NE independently examined the first five transcripts to develop a coding framework which was applied to the analysis of the remaining transcripts. Interview data were categorised and compared enabling identification of common themes and sub-themes. An initial list of themes and sub-themes were created, and then examined for themes containing few quotations or overlapping to be merged under other major themes or sub-themes. Throughout this process, the data were consistently analysed to gain insight into the relationship between themes. The themes were discussed among researchers to improve the confirmability, dependability, and trustworthiness of the findings. The findings were interpreted within the context of the existing literature.

## Inclusivity in global research

The research team maintained integrity, participants' privacy, and confidentiality throughout the study. Project members from both the UK and China met regularly to monitor recruitment, review the progress of the study, and make recommendations for the overall direction and conduct of the study. Additional information regarding the ethical, cultural, and scientific considerations specific to inclusivity in global research is included in the S1 Data.

## Findings

### Participants' characteristics

A total of 26 healthcare workers were approached, of which 20 (76.9%) agreed to participate in the qualitative study (Table 1). Thirteen (65.0%) were nurses and seven (35.0%) were physicians. Interviews lasted between 20 to 40 minutes (Mean = 25 minutes). Participants had an average age of 35.9 years (range = 25–52), with an average working experience of 12.8 years (range = 3–25). They were redeployed to COVID-19 isolation wards in Wuhan from January 26 to March 25, 2020 (60 days).

Four themes were created from the analysis of the interviews: (1) Initial feelings and emotions of redeployment, (2) 'It is like a war zone', (3) Uncertainty and coping strategies in patient care, and (4) Reflection and far-reaching impacts of caring for COVID-19 patients (Table 2).

### Theme 1 Initial feelings and emotions of redeployment

Along with healthcare workers from the whole country rallying and redeploying to the epicentre of COVID-19 during the early phase of the outbreak, physicians and nurses in Henan were also mobilised to Wuhan hospitals to support local staff in caring for COVID-19 patients.

**Making difficult decisions to go to the epicentre.** The majority of participants felt concerned and shocked when they were asked whether they would be willing to go to Wuhan by head nurses or head of departments on the day before the Spring Festival. However, they knew that they had to make quick decisions and hoped to get support from their loved ones.

*"I was really hesitant and struggled at that time because I didn't know anything about the situation. I didn't know what was going on, and suddenly we were assembled to Wuhan."* (Physician 6, age 38, female)

*"I was worried about my family; my little baby was only four or five months old. I was hesitant, my wife also did not want me to go. I hesitated for a few minutes. But four or five*

**Table 1. Characteristics of healthcare workers redeployed to Wuhan (n = 20).**

| | |
|---|---|
| **Gender** | |
| Female | 11 |
| Male | 9 |
| **Role** | |
| Physician | 7 |
| Nurse | 13 |
| **Age range** | |
| 20–29 | 4 |
| 30–39 | 12 |
| 40–49 | 2 |
| 50–59 | 2 |
| **Marital status** | |
| Single | 5 |
| Married | 15 |
| **Education** | |
| Diploma | 2 |
| Bachelor | 14 |
| Master | 4 |
| **Work experience, years** | |
| 0–4 | 1 |
| 5–9 | 5 |
| 10–14 | 7 |
| 15–19 | 3 |
| 20–24 | 2 |
| 25–29 | 2 |
| **Original department** | |
| Infectious disease | 3 |
| Intensive Care Unit (ICU) | 7 |
| Emergency | 2 |
| Respiratory | 8 |

**Table 2. Themes and subthemes identified from interviews.**

| Themes | Subthemes |
|---|---|
| Initial feelings and emotions of redeployment | Making difficult decisions to go to the epicentre |
| | Emotions in the early stage of arrival |
| 'It is like a war zone' | Lack of resources and knowledge |
| | Challenges of delivering care in an unfamiliar environment |
| Uncertainty and coping strategies in patient care | Psychological care and palliative care |
| | Virtual communication |
| | Relationship between patients and staff |
| Reflection and far-reaching impacts of caring for COVID-19 patients | Heroism, patriotism, and willingness to sacrifice |
| | Feeling motivated and encouraged |

*minutes later I made up my mind [to go to Wuhan and offer support]."* (Physician 7, age 37, male)

*"My husband was very supportive, particularly at these critical moments, and he didn't hold me back. He had encouraged me to go and reassured me that I didn't need to worry about things at home and he would take care of everything. Anyway, his support made me feel relieved and pleased."* (Nurse 1, age 34, female)

**Emotions in the early stage of arrival.**    A few participants expressed their worries and concerns, sense of isolation and loneliness upon arrival to the epicentre. As a nurse described,

*"When we got off the train, the station was empty, and it was late and dark. Two coaches sent us to the hotel. It was really an empty city. There was nothing, just empty streets and buildings. We were silent during the coach trip. We must be so scared and frightened, but no one said anything."* (Nurse 1, age 34, female)

Participants reported they had worries and concerns arising from the pandemic due to the uncertainty of the disease in the early stage. A physician said,

*"We didn't know about the virus. I had no confidence in fighting against it. It was not like the common cold. We knew how to treat a viral or bacterial flu. But we didn't know much about COVID-19. We didn't even know how contagious it would be."* (Physician 3, age 47, male)

During the first four weeks, participants were restricted in their hotel rooms after their shifts because of the lockdown policies. Some participants felt very lonely.

*"When we arrived at the hotel, we were told that we would not be allowed to visit each other, particularly when we finish our shift to control and minimize possible contamination. I felt miserable and lonely without being able to talk and interact with people. When I woke up at night, I couldn't walk around and couldn't talk with anyone. I just looked at the building behind our hotel. I felt very lonely."* (Nurse 4, age 39, male)

## Theme 2 'It was like a war zone'

Healthcare workers talked about their daily challenges against COVID-19 with one nurse reporting that "*it was like a war zone*" (Nurse 4, age 39, male). These frontline workers shared the risks and challenges they faced, and despite the hardships they experienced, they continued to work towards fighting against COVID-19 in Wuhan and valued the opportunities "*to reflect what had been done well and what could be improved or done differently, and to make necessary adjustments accordingly.*" (Physician 3, age 47, male).

**Lack of resources and knowledge.**    Participants reported the challenges of being overstretched and lacking resources to combat the crisis. They expressed concerns about ventilators and PPE stockpiles running low and inappropriate equipment donated by the public. They had to "*limit the number of staff entering the isolation wards to save PPE and protect the healthcare workers*". (Physician 2, age 44, female).

*"In two weeks or so, the resources were short of supply including PPE and N95 masks. So we decreased the number of healthcare workers in the isolation wards to save resources. We were not allowed to go into the isolation wards without good protection. Some severely ill patients*

*had to be intubated. But we didn't have enough endotracheal tubes to use in the hospital."* (Nurse 5, age 34 female)

Hospitals had to prioritise patients for admission due to the shortage of beds.

*"I was deeply impressed by an old man. He and his son got infected. But the hospital was full, so his son had to stay home for isolation. He kept asking us to let him out, and let his son in."* (Nurse 8, age 29, female)

A lack of adequate knowledge about COVID-19 hindered medical and care provision. Confusion and tension were present about mortality rates, disease prognosis and trajectory, effective therapies, and patient outcomes, particularly in the early stages of the outbreak. As a physician explained,

*"At the early stage, we didn't have much knowledge about COVID-19 protection and treatment, neither about the severity of the pandemic in Wuhan. All of us were probably so nervous when we got there. My mind went blank."* (Physician 5, age 50, male)

*"When I got back to the hotel after work, I took a shower, then grabbed my phone and searched for information about COVID-19. National guidelines were released in a short time. We have learnt from it every day, not only about the knowledge of COVID-19, but also hypoxia, nutritional support, because it is a complex disease. I had to relearn them."* (Physician 3, age 47, male)

**Challenges of delivering care in an unfamiliar environment.** Due to the highly infectious nature of COVID-19 and a large number of admitted patients with COVID-19, some of the normal wards were redesigned to mitigate cross-contamination risks. However, healthcare workers had to respond to and manage this crisis as quickly as they could and start caring for the patients under huge pressure before the completion of reconstruction. A physician reported that,

*"The hospital where we were redeployed was originally an orthopaedic hospital. Wards were reconstructed temporarily. According to the standards of infectious disease ward, they should at least have three areas and two channels, which are clean area, contaminated area, semi-contaminated area, patient routes and staff routes. But this hospital didn't have them."* (Physician 3, age 47, male)

*"The next day, a head nurse and physicians from the infectious disease department came and helped to reorganise the ward. Because the current wards were not suitable for infectious patients. We had to help to redesign the current wards to become an isolation ward which was relatively acceptable for caring for patients with infectious diseases."* (Nurse 1, age 34, female)

Language barriers added another layer of complexity to the crisis, and miscommunication was identified as one of main challenges which influenced the delivery of healthcare and patient satisfaction as the participants "could not speak and understand the local dialect" (e.g., Nurse 10, age 32, female). They often "had to write it down or ask the local staff to help to translate" for them (e.g., Physician 1, age 39, male; Nurse 6, age 28, female).

The process of redeployment can be a difficult experience for staff, as working in new environments with new systems and unfamiliar colleagues can all affect morale. Routine work

(e.g., dealing with medical orders) became difficult as healthcare workers had to learn the local electronic system (Nurse 10, age 32, female). Staff established professional collegiality for supporting each other, coordinating care during uncertainty, and ensuring quality and safety.

> *"Because of the new environment, the new work process was developed immediately, but you still need time to implement, need people to learn, to run it. Every time we went into the isolation ward, we had to check out the working process very carefully, resolve problems coming from the process until everything went on very smoothly."* (Physician 6, age 38, female)

Some participants commented that wearing PPE was extremely tiring and uncomfortable, which made it more difficult to deliver care.

> *"Wearing protective clothing is inconvenient to walk. I also could not drink or go to the bathroom because the protective clothing was very expensive, so I would not drink before going into the isolation wards. Our colleagues who had gastrointestinal problems would eat less or not eat at all."* (Physician 2, age 44, female)

> *"When I was wearing double layer gloves, I could not feel the blood vessels, especially for patients whose vein was not palpable."* (Nurse 1, age 34, female)

### Theme 3 Uncertainties and coping strategies in patient care

**Psychological care and palliative care.**    The negative impacts of COVID-19 had rippled through every facet of patients' life and caused psychological concerns. Participants witnessed patients developing post-traumatic stress disorder, anxiety, depression, and other symptoms of distress, and valued the importance of patient-centred care. As participants explained,

> *"Some patients felt that they had no hope of living. Some wanted to commit suicide. They felt it was incurable and they had a terminal disease. If they lost hope, they would die."* (Physician 2, age 44, female)

> *"I think they were in a desperate state. When I got there, the patient was desperate. They had needs, and they did not speak them out. We did not know what they had experienced, but they didn't communicate much with us. . ."* (Nurse 1, age34, female)

> *"Before we arrived, the mortality rate was very high. Patients felt that they were going to die. Their mood was very low. I went into the wards, talked with them and gave them hope. Gradually they had built up faith in survival. We asked them to eat to gain energy even though they didn't want to."* (Physician 3, age 47, male).

Participants emphasised that many patients with severe COVID-19 experienced distressing symptoms, including breathlessness and agitation, and they actively cooperated with the staff treating them and appreciated the care received at their end of life. Palliation of suffering was considered an important part of care irrespective of prognosis. Some staff had some knowledge about palliative care and experience of taking care of end-of-life patients while others demonstrated misunderstanding of palliative care.

> *"On one of my night shifts, an old lady aged more than 80 with critical illness died. Before the night she died. . . when we helped her to change her incontinence pad and turn over, she said "Thank you". Afterwards, I didn't hear her say anything because she was critically ill and*

*could barely speak. The "Thank you" she said that night touched me very much."* (Physician 7, age 37, male)

*"We would not give up until the last minutes. The treatment and care would not end until the patient is dead. If there's a heartbeat or a breath, we won't think they're hopeless and give them up."* (Physician 4, age 52, male)

*"From a medical point of view, continuing treatment for those who are at the very end of life may be expensive for very little or no benefit. In addition, it might be against the patients' will. But there is not much we can do as it may be hugely affected by the traditional beliefs and culture in our country. In most circumstances, we tended to listen to what their family members suggested. If the family members wanted to continue treatment, we will give them positive treatment. Even if the patient might not want to, we would still respect and act upon the decisions made by family members."* (Physician 1, age 39, male)

**Virtual communication.**   Amid the COVID-19 pandemic, most communication and information sharing between healthcare workers, patients and their families were online. Virtual communication has been embraced as never before. Healthcare workers called family members for discussing their preferences and treatment strategies for their loved ones. Besides, participants reported how patients contacted their families virtually while in the isolation wards.

*"Family members cannot accompany the patient due to the requirement of these isolation wards. For example, a family of four, one was isolated at this hospital, one was at another hospital. Almost the entire family was under quarantine. Some of them have died, but their family members didn't even know it. Afterwards, they might communicate via the telephone. The early phase was completely isolated."* (Nurse 1, age 34, female)

**Relationship between patients and staff.**   Building mutual trust between patients and staff was reported as key to success in treating patients with COVID-19. Participants found that patients were dependent on staff and appreciated communicating with them.

*"A friend in need is a friend indeed. After we went there, we entrusted our lives to patients while they entrusted theirs to us. We were like comrades in arms. In this war, we had a common goal to fight against the epidemic."* (Nurse 7, age 25, female)

However, some participants commented that such a harmonious patient and staff relationship could be temporary, and it might get back to normal immediately after the pandemic.

*"When the COVID-19 was under control, patients may have a higher demand of needs. If you met their basic needs, they would be fine. For example, at the early stage, all the hospitals in Wuhan were short of medical staff. When we got there, we treated and cared for them, we saved their lives. They really appreciated it. When they got better, they would want to go outside, they would want freedom. They would want more."* (Nurse 6, age 28, female)

## Theme 4 Reflection and far-reaching impacts of caring for COVID-19 patients

**Heroism, patriotism, and willingness to sacrifice.**   Heroism, patriotism, and willingness to sacrifice has been fully displayed by the participants and it is a spiritual hallmark of the

Chinese nation and a strong bond uniting Chinese people. All participants showed their professionalism and fully understood what scarifying of individual interests for collective benefits means.

*"If you have an opportunity to help your country, you will take it without thinking. This time, when our country and people need us, we should be there as professional healthcare staff members."* (Nurse 4, age 39, male)

*"After we came back, most colleagues, relatives, and citizens called us heroes. It was not a big deal. I just did my job."* (Nurse 6, age 28, female)

Participants felt that they and their families were supported by the country and everyone around them, including colleagues from their own hospitals and hotel staff in Wuhan, and developed a sense of achievement.

*"After coming back, the government and the hospital gave a lot of honours to us. I have never felt so proud since I started medicine training 20 years ago. I felt proud to be a physician."* (Physician 1, age 39, male)

*"The country did a good job in supporting my family members back at home. I felt very grateful."* (Nurse 7, age 25, female)

**Feeling motivated and encouraged.** As the redeployed healthcare workers came back to their original workplace, they reflected on what they had experienced in the epicentre of the COVID-19 pandemic. Most participants indicated that following the redeployment experience they would cherish every moment they spent with their loved ones, and they could understand patients and their family much better.

*"Every time we came out of the isolation ward, we joked with each other that we had survived another day. Everyone knew we were lucky to get out alive. Under certain circumstances, you will find that your health is most important. Today we can be with our families and hold our children."* (Nurse 1, age 34, female)

*"I cherish my life, and every moment spent with relatives, friends and the peaceful life now. Happy life doesn't come easily, everyone has made extraordinary efforts."* (Physician 7, age 37, male)

*"I could understand patients better. Because in Wuhan, everything had to be done by staff without any help from family members. To be frank I can understand the hardship of families and caregivers."* (Nurse 10, age 32, female)

Some participants also expressed that being redeployed to Wuhan was an invaluable experience which made them feel stronger, more confident, and motivated at work.

*"From a personal perspective, my psychological wellbeing has been improved. This was an invaluable experience. My life views became more practical and pragmatic. I am more accepting of everything in life. I would not mind many trivial things in family and work, which I would mind before."* (Physician 6, age 38, female)

*"I have worked for many years, but I have not been praised for my work. After Wuhan's trip, the patients gave you a lot of confidence and encouragement, which was the recognition of my worth. I feel more passionate about my work."* (Nurse 3, age 36, male)

## Discussion

This study confirms that redeployed healthcare workers were exposed to a range of extreme conditions associated with the COVID-19 pandemic and experienced challenges while caring for COVID-19 patients in the context of resource strain in Wuhan. Our findings showed that redeployment, lack of sufficient preparedness and training, support, knowledge, and excessive workload gave rise to psychological burden. At the beginning of the pandemic, staff experienced anxiety, and had to balance the needs of their own families against the demand of their jobs. However, this dissipated and was replaced by increased senses of professionalism and patriotism. Over time, knowledge around the COVID-19 infection, healthcare workers' skills, and the working environment improved. With the containment of the initial wave of the pandemic, healthcare workers felt motivated, encouraged, and rewarded as efforts were recognised by hospital managers as well as the government and wider society.

A general sense of anxiety and uncertainty in the early stages of the outbreak was expressed by the participants in this study regarding the pandemic and redeployment experience, that placed participants and their organizations in uncharted territory. Systematic reviews have highlighted the burden of mental health symptoms including anxiety, acute stress, depression, and burnout among frontline healthcare workers during and following a disease outbreak [39, 40]. Participants in our study experienced varying degrees of anxiety, particularly with respect to concerns for their families. Other qualitative studies also reported that healthcare workers experienced anxiety about working in the COVID-19 wards and becoming infected themselves [41, 42].

Since the outbreak of SARS in 2003, infectious diseases have posed significant psychological threats to healthcare workers [43, 44]. A systematic review has indicated that the psychological impact on staff appears to be associated with occupational role, training and preparedness, high-risk work environments, quarantine, role-related stressors, perceived risk, social support, social rejection and isolation, and personal or professional life [45]. Factors which are associated with less psychological distress include personal and organizational social support, perceived control, positive work attitudes, sufficient information about the outbreak and proper protection, training, and resources [46].

Participants in this study were redeployed and asked to temporarily work in a different or unfamiliar care setting to support the ongoing pandemic response. Evidence on the experience and readiness of redeployment in a pandemic and the barriers and facilitators of effective redeployment remain unclear [47]. Studies have found leadership, information provision and access to workplace resources as mediating factors in improving emergency preparedness among redeployed nurses [25, 48, 49]. The issue of how best to support redeployed healthcare workers has attracted extensive attention worldwide [50–53]. NHS England and NHS Improvement published national guidance on key considerations for the safe redeployment of staff and deployment of those joining the NHS temporarily to support the existing workforce [54]. United States Centres for Disease Control and Prevention provided guidance for redeployed teams on capacity development at the national and sub-national levels during an active COVID-19 response [55]. However, there was no guidance to support redeployed healthcare workers in China. It is crucial to recognise that they will need sufficient measures in place to support this unique transition in order to provide the best care for patients and ensure a safe working environment [56, 57].

Caring for critically ill patients with a known or suspected novel infectious disease during a global pandemic is a complex task and often requires palliative care knowledge and skills to ensure provision of holistic and compassionate care. Providing effective palliative care is an essential and vital component of care for severely ill COVID-19 patients, but it has proved to

be difficult in the context of a pandemic [1, 58]. Some participants in our study tended to misunderstand the concept of palliative care and were less likely to initiate palliative care for COVID-19 patients. Some staff had a misconception that once they initiated palliative care, it meant that they were 'giving up' on the patient and leaving them waiting to die. Cultural factors play an important role in the delivery of palliative care [59]. Our findings were consistent with previous studies from Chinese cultural contexts [60–63]. Talking about death and dying and giving up treatment is very challenging for staff in China [64, 65]. Discussions about death are often avoided as they are seen hastening the pace of patients' dying process and incurring bad luck [66, 67]. Indeed, palliative care is not emphasized enough in national policy or clinical practice guidelines for COVID-19 in China [68]. Correcting this inaccurate understanding of palliative care and reaffirming the need for effective palliative care training for healthcare workers at a national level is crucial to ensure holistic care for COVID-19 patients in China and other countries with similar misconceptions about palliative care.

The availability of resources is clearly an important mediator in successfully managing the care of critically ill patients in pandemics [69]. Research during previous infectious disease outbreaks has shown that ample supplies of PPE significantly facilitated effective clinical care, while insufficient or rapidly depleting PPE contributed to healthcare workers' anxiety [70]. When considering the management of essential resources required to care for patients with COVID-19, inadequate supplies of equipment (i.e., ventilators, isolation wards) and possible PPE shortages were particularly concerning [71, 72]. Our study participants expressed similar concerns. Some participants described that their units had developed and enacted detailed plans to prepare for surge scenarios, including a tiered response of staffing and capacity depending on the number of patients with COVID-19 admitted to the hospital. Other participants felt that their units were less prepared because they were already experiencing shortages of PPE and conserving remaining supplies. In addition to echoing the importance of adequate material resources, several of our participants also stressed the need for COVID-19 related training. In this respect, participants reported the lack of adequate knowledge about COVID-19 and their particular concern which was working in an unfamiliar environment at reconstructed facilities in the epicentre.

In the COVID-19 setting, healthcare workers were praised for their impossible efforts in caring for those in need. Our study participants showed willingness to sacrifice in the context of resource strain, belief in the ability to triumph over adversity, patriotism and faith involving love for their country. Professional commitment and patriotism were important factors affecting frontline healthcare workers' willingness to work during a pandemic, which has significant implications for maintaining workforce stability and quality of care at a time of elevated health needs [73]. Being sustained by their belief in the ability to triumph over adversity, healthcare workers were more confident and willing to engage in frontline work during the COVID-19 pandemic [74]. Their professional values and self-fulfilment were boosted by appreciation from patients and society [75].

Participants in our study appreciated the appraisal of their efforts by the society, however, there is an increasing concern about the increasing visibility in social media, and terms such as "heroes" that were used to describe healthcare workers can have potentially negative consequences [76, 77]. The overuse of the concept of heroism in the media could have a negative psychological impact, through the expectation that all healthcare workers must act in a heroic manner [78]. It is unreasonable to demand heroism as the norm and no one is obliged to perform supererogatory acts [79]. Rather than praising all healthcare workers as heroes, we need to shift from heroism to humility [80]. It is important to critically examine what reasonable duties they have to perform and how they can be reciprocally supported [78].

The reality of a pandemic response may reduce healthcare workers' willingness to work due to various reasons including inadequate preparedness of facility and workplace safety [81].

Absenteeism, although not mentioned by the participants in our study, can be a challenge during a pandemic and can be caused by fear of infection, duties to one's family and loved ones, lack of resources and trust in the management of facilities and government [82]. It is important to have in place effective crisis management plans in relation to human resources and provide relevant training to ensure adequate staff are available to work during pandemics [83]. Theories of self-determination [84] and intrinsic and extrinsic incentives [85] could be useful approaches to promoting motivation and performance of healthcare workers in the crisis [86–91].

Maintaining an adequate healthcare workforce in this crisis requires not only an adequate number of physicians, nurses, advanced practice clinicians, pharmacists, respiratory therapists, and other clinicians, but also identifying strategies to maximize the ability of each clinician to care for a high volume of patients [44]. Given that surges in critically ill patients could last weeks to months, it is also essential that healthcare workers are supported by their organizations to sustain performance to full potential over extended periods. In order to effectively support frontline healthcare workers and address the challenges they are facing during the pandemic, it is critical to continue investigating their experiences and needs in the fight against COVID-19 in China and across the world [25].

One of the strengths of our study was rigorous data analysis, with all steps taken to maximize the validity and trustworthiness of the findings. We deliberately sought to include participants with different backgrounds, including physicians and nurses, to explore the diversity of experiences during the pandemic and increase the potential transferability of our findings. Our research team members included diverse clinical and research expertise, which significantly improved rigorous recruitment and data collection process, and enhanced the interpretation of the interview data and report of our study findings. In addition, the timing of the interviews meant that participants had sufficient time to reflect on their experience after redeployment.

Nevertheless, this study has some limitations. Despite our efforts to include participants from diverse backgrounds, we were only able to recruit participants who were redeployed to Wuhan from one study site in Henan. The experiences and needs of a wider range of healthcare workers who were redeployed to Wuhan from multiple sites or who were redeployed to other locations would need to be explored in future qualitative studies to enrich our understanding of redeployment experiences during the pandemic.

## Conclusions

This qualitative study comprehensively demonstrated how redeployed healthcare workers in the initial epicentre of COVID-19 in China were impacted by being on the front line of a pandemic. They shared unique experiences and a range of challenges in the context of resource strain which create significant implications for policy and future practice. The themes constructed in this study can be used by managers and policy makers to mitigate negative psychological outcomes of staff and promote emergency and logistic preparedness and response. Continuously evolving pandemic circumstances and a sense of uncertainty expressed by our participants highlight the importance of an organized national pandemic response plan for subsequent waves of COVID-19 and future pandemics.

## Supporting information

**S1 File. Topic guide for staff interview participants.** Understanding your needs and experiences of caring COVID-19 patients.
(DOCX)

**S1 Data. Inclusivity in global research.**
(DOCX)

# Acknowledgments

Our sincere thanks go to all participants for their commitment. They substantially contributed to this work through sharing their unique experiences and significant insights on how to cope with redeployment and challenges faced while caring for patients with COVID-19 in Wuhan, China. This study was supported by the School of Nursing and Health, Henan University.

# Author Contributions

**Conceptualization:** Nikolaos Efstathiou, Ping Guo.

**Data curation:** Houshen Li, Yifan Cui, Nikolaos Efstathiou, Bo Li, Ping Guo.

**Formal analysis:** Houshen Li, Yifan Cui, Nikolaos Efstathiou, Bo Li, Ping Guo.

**Investigation:** Houshen Li, Yifan Cui, Nikolaos Efstathiou, Bo Li, Ping Guo.

**Methodology:** Nikolaos Efstathiou, Bo Li, Ping Guo.

**Supervision:** Nikolaos Efstathiou, Bo Li, Ping Guo.

**Writing – original draft:** Houshen Li, Yifan Cui, Ping Guo.

**Writing – review & editing:** Houshen Li, Yifan Cui, Nikolaos Efstathiou, Bo Li, Ping Guo.

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
