## [Decision Letter · Decision Letter 0]

22 Feb 2022

PONE-D-21-40460Experiences and needs of redeployed frontline healthcare providers in the fight against COVID-19 in China: a qualitative interview studyPLOS ONE

Dear Dr. Guo,

Thank you for submitting your manuscript to PLOS ONE. After careful consideration, we feel that it has merit but does not fully meet PLOS ONE’s publication criteria as it currently stands. Therefore, we invite you to submit a revised version of the manuscript that addresses the points raised during the review process.

We look forward to receiving your revised manuscript.

Kind regards,

Xin Liu, Ph.D.

Academic Editor

PLOS ONE

Journal Requirements:

Additional Editor Comments:

The research is of great significance in the fight against COVID-19 in China. But paper structure including introduction, discussion, and conclusion sections, is confused and should be improved. In the method section, more details of sample, data, variables, and data statistics are needed in the revised version. Theoretical framework is weak and the previous relevant COVID-19 references should be cited to to provide better implications of management of healthcare.

Reviewers' comments:

Reviewer's Responses to Questions

**Comments to the Author**

1. Is the manuscript technically sound, and do the data support the conclusions?

Reviewer #1: No

Reviewer #2: Partly

Reviewer #3: Partly

2. Has the statistical analysis been performed appropriately and rigorously? 

Reviewer #1: No

Reviewer #2: N/A

Reviewer #3: No

3. Have the authors made all data underlying the findings in their manuscript fully available?

Reviewer #1: No

Reviewer #2: Yes

Reviewer #3: Yes

4. Is the manuscript presented in an intelligible fashion and written in standard English?

Reviewer #1: Yes

Reviewer #2: Yes

Reviewer #3: Yes

5. Review Comments to the Author

Reviewer #1: 1. As the author mentioned that the participants attended COVID-19 isolation wards in Wuhan from 26th January to 26th March 2020 (60 days), but the Face-to-face, semi-structured in-depth interviews were conducted 1st September 2020 and 31st May 2021. There had a long time interval, which may impact the feeling of the participants.

2. The introduction should rewritten, which should describe the purpose and significance of this study.

3. The results of the main found should be expressed using tables, the format of present was not suitable for a paper.

4. The discussion should focus on the main new discovery of this manuscript, and compared with other previous studies.

Reviewer #2: Experiences and needs of redeployed frontline healthcare providers in the fight against COVID-19 in China: a qualitative interview study

The topics of this paper are interesting, but the structure and content must be revised, and results have to be better explained by authors before to be reconsidered for publication.

Title has to be shorter and fit better the topics under study.

Abstract has to be shorter focusing on results, and health and social implications of this study for improving healthcare management.

Introduction has to better clarify the research questions of this study and provide more theoretical background. Authors have to better describe the different sources of transmission dynamics of COVID-19 and risk factors in society, as well as effects of the policy of lockdown on organizations and society in China (See suggested readings that must be all read and used in the text).

Methods of this study is not clear. The section of Materials and methods must be re-structured with following three sections:

• Sample and data

• Measures of variables

• Data analysis procedure.

Study design can be improved.

Current sections can be reorganized in the just mentioned sections.

Results are confused.

Although the study is qualitative, I suggest synthetizing and systematize results in tables showing main findings according to the categorization of working place/department (Emergency, ICU, Respiratory and infectious disease), marital status and gender to detect similarity and/or differences of effects on working condition and mental health to provide better implications of management of healthcare.

As scholars this is the research activity to do with this qualitative information to extend the knowledge in these topics. Results with these table can be synthetized better instead of presenting all sentences.

Discussion can remove sub-headings that create fragmentation and confusion. Discussion to be fruitful and useful to cope with future pandemics have to support implications of crisis management of human resource, using theories of self-determination, intrinsic and extrinsic incentives to reduce negative effects in organizations in problematic situation (suggested papers can support a similar and fruitful discussion) to increase the impact of this paper and utility for readers of the journal, such as managers and policymakers.

Discussion should also consider similar problems in other countries (see suggested papers) .

Conclusion is poor and must be rewritten and extended. It has not to be a summary, but authors must focus on manifold limitations of this study and provide suggestions of health, crisis management and social policy. As well as strategies of preparedness and prevention of future pandemic diseases (see suggested docs).

Overall, then, the paper is interesting, but structure is confused. Theoretical framework is weak, and some results create confusion… structure of the paper must be improved; study design, discussion and presentation of results have to be clarified using suggested comments .

If the paper is improved, by using all comments, maybe it can be considered.

Suggested readings of relevant papers that must be read, and all inserted in the text and references.

Blanco-Donoso, L.M., Moreno-Jiménez, J., Gallego-Alberto, L., (...), Moreno-Jiménez, B., Garrosa, E. 2022Satisfied as professionals, but also exhausted and worried!!: The role of job demands, resources and emotional experiences of Spanish nursing home workers during the COVID-19 pandemic, Health and Social Care in the Community30(1), pp. e148-e160

Coccia M. 2021. The relation between length of lockdown, numbers of infected people and deaths of Covid-19, and economic growth of countries: Lessons learned to cope with future pandemics similar to Covid-19. Science of The Total Environment, n. 145801. https://doi.org/10.1016/j.scitotenv.2021.145801

Pereira, A.T., Cabaços, C., Araújo, A., (...), Carvalho, F., Macedo, A. 2022. COVID-19 psychological impact: The role of perfectionism, Personality and Individual Differences 184,111160

Zhang, C., Wang, C., Chen, C., (...), Wang, Z., Jia, B. 2022. Effects of tree canopy on psychological distress: A repeated cross-sectional study before and during the COVID-19 epidemic, Environmental Research 203,111795

Coccia M. 2020. How (Un)sustainable Environments are Related to the Diffusion of COVID-19: The Relation between Coronavirus Disease 2019, Air Pollution, Wind Resource and Energy. Sustainability 2020, 12, 9709; doi:10.3390/su12229709

Andersen, A.J., Mary-Krause, M., Bustamante, J.J.H., (...), El Aarbaoui, T., Melchior, M. 2021. Symptoms of anxiety/depression during the COVID-19 pandemic and associated lockdown in the community: longitudinal data from the TEMPO cohort in France, BMC Psychiatry21(1),381

Coccia M. 2021. Effects of the spread of COVID-19 on public health of polluted cities: results of the first wave for explaining the dejà vu in the second wave of COVID-19 pandemic and epidemics of future vital agents. Environmental Science and Pollution Research. 28(15), 19147-19154. https://doi.org/10.1007/s11356-020-11662-7

Salari, N., Hosseinian-Far, A., Jalali, R., Vaisi-Raygani, A., Rasoulpoor, S., Mohammadi, M., Rasoulpoor, S., (...), Khaledi-Paveh, B. 2020.Prevalence of stress, anxiety, depression among the general population during the COVID-19 pandemic: A systematic review and meta-analysis (Open Access), (2020) Globalization and Health, 16 (1), art. no. 57

Al Hariri, M., Hamade, B., Bizri, M., (...), Tamim, H., Al Jalbout, N. 2022 Psychological impact of COVID-19 on emergency department healthcare workers in a tertiary care center during a national economic crisis, American Journal of Emergency Medicine51, pp. 342-347

Coccia M. 2021. Pandemic Prevention: Lessons from COVID-19. Encyclopedia of COVID-19, 1, pp. 433–444. https://doi.org/10.3390/encyclopedia1020036

Trepanier, S., Henderson, R., Waghray, A. 2022A health care system's approach to support nursing leaders in mitigating burnout amid a COVID-19 world pandemic Nursing Administration Quarterly46(1), pp. 52-59

Xiang, Y.-T., Jin, Y., Cheung, T. 2020. Joint International Collaboration to Combat Mental Health Challenges during the Coronavirus Disease 2019 Pandemic, (2020) JAMA Psychiatry, 77 (10), pp. 989-990.

Coccia M. 2021. High health expenditures and low exposure of population to air pollution as critical factors that can reduce fatality rate in COVID-19 pandemic crisis: a global analysis. Environmental Research, vol. 199, Article number 111339, https://doi.org/10.1016/j.envres.2021.111339

Dopelt, K., Bashkin, O., Davidovitch, N., Asna, N. 2021. Facing the unknown: Healthcare workers’ concerns, experiences, and burnout during the covid-19 pandemic— a mixed-methods study in an Israeli hospital, Sustainability (Switzerland)13(16),9021

Feingold, J.H., Hurtado, A., Feder, A., (...), Ripp, J., Pietrzak, R.H. 2022Posttraumatic growth among health care workers on the frontlines of the COVID-19 pandemic, Journal of Affective Disorders296, pp. 35-40

Coccia M. 2019. Theories of Self-determination. A. Farazmand (ed.), Global Encyclopedia of Public Administration, Public Policy, and Governance, Springer Nature Switzerland AG, https://doi.org/10.1007/978-3-319-31816-5_3710-1

Azizkhani, R., Heydari, F., Sadeghi, A., Ahmadi, O., Meibody, A.A. 2022. Professional quality of life and emotional well-being among healthcare workers during the COVID-19 pandemic in Iran, Frontiers in Emergency Medicine6(1),e2

Coccia M. 2018. Motivation and theory of self-determination: Some management implications in organizations, Journal of Economics Bibliography, vol. 5, n. 4, pp. 223-230, http://dx.doi.org/10.1453/jeb.v5i4.1792

Butler, C. R., Wong, S., Vig, E. K., Neely, C. S., & O'Hare, A. M. (2021). Professional roles and relationships during the COVID-19 pandemic: a qualitative study among US clinicians. BMJ open, 11(3), e047782. https://doi.org/10.1136/bmjopen-2020-047782

Coccia M. 2019. Comparative Incentive Systems. A. Farazmand (ed.), Global Encyclopedia of Public Administration, Public Policy, and Governance, Springer Nature Switzerland AG, https://doi.org/10.1007/978-3-319-31816-5_3706-1.

Mensinger, J.L., Brom, H., Havens, D.S., (...), Yost, J., Kaufmann, P. 2022Psychological responses of hospital-based nurses working during the COVID-19 pandemic in the United States: A cross-sectional study, Applied Nursing Research63,151517

Coccia M. 2019. Intrinsic and extrinsic incentives to support motivation and performance of public organizations, Journal of Economics Bibliography, vol. 6, no. 1, pp. 20-29, http://dx.doi.org/10.1453/jeb.v6i1.1795,

de Medeiros, K.S., de Paiva, L.M.F., de Araújo Macêdo, L.T., (...), Freitas, C.L., Gonçalves, A.K. 2021Prevalence of Burnout Syndrome and other psychiatric disorders among health professionals during the COVID-19 pandemic: A systematic review and meta-analysis protocol, PLoS ONE 16(12 December),e0260410

Reviewer #3: The aim of this study was to explore the experiences and needs of frontline healthcare workers who were redeployed

to care for COVID-19 patients in Wuhan, China. This is a well-written and well-presented article on an important topic. This research is qualitative and shows no statistical data. Therefore, I would expect to see a better conclusion at the end of the manuscript.

6. PLOS authors have the option to publish the peer review history of their article (what does this mean?). If published, this will include your full peer review and any attached files.

Reviewer #1: **Yes: **Xiaohua Liang

Reviewer #2: No

Reviewer #3: No

---

## [Author Response · Author response to Decision Letter 0]

26 Apr 2022

A document named 'Response to reviewer PLOS ONE R1 08.04.22' has been uploaded as a file with the manuscript and other documents.

---

## [Decision Letter · Decision Letter 1]

19 Jul 2022

PONE-D-21-40460R1Experiences of redeployed healthcare providers in the fight against COVID-19 in China: a qualitative studyPLOS ONE

Dear Dr. Guo,

Thank you for submitting your manuscript to PLOS ONE. After careful consideration, we feel that it has merit but does not fully meet PLOS ONE’s publication criteria as it currently stands. Therefore, we invite you to submit a revised version of the manuscript that addresses the points raised during the review process. We recommend that you either get your manuscript reviewed by someone who is fluent in English or, if you would like professional help, you can use any reputable English language editing service.

We look forward to receiving your revised manuscript.

Kind regards,

Xin Liu, Ph.D.

Academic Editor

PLOS ONE

Journal Requirements:

Reviewers' comments:

Reviewer's Responses to Questions

**Comments to the Author**

1. If the authors have adequately addressed your comments raised in a previous round of review and you feel that this manuscript is now acceptable for publication, you may indicate that here to bypass the “Comments to the Author” section, enter your conflict of interest statement in the “Confidential to Editor” section, and submit your "Accept" recommendation.

Reviewer #2: All comments have been addressed

Reviewer #3: All comments have been addressed

2. Is the manuscript technically sound, and do the data support the conclusions?

Reviewer #2: Yes

Reviewer #3: Yes

3. Has the statistical analysis been performed appropriately and rigorously? 

Reviewer #2: Yes

Reviewer #3: N/A

4. Have the authors made all data underlying the findings in their manuscript fully available?

Reviewer #2: Yes

Reviewer #3: Yes

5. Is the manuscript presented in an intelligible fashion and written in standard English?

Reviewer #2: Yes

Reviewer #3: Yes

6. Review Comments to the Author

Reviewer #2: I have read thoroughly the revised version of paper.

Now this version of the paper after revision done is OK and provides interesting results for readers of PLOS ONE.

Reviewer #3: The authors have successfully addressed the comments. The manuscript can be considered for publication.

7. PLOS authors have the option to publish the peer review history of their article (what does this mean?). If published, this will include your full peer review and any attached files.

Reviewer #2: No

Reviewer #3: No

---

## [Author Response · Author response to Decision Letter 1]

6 Aug 2022

Thank you so much for your review and helpful comments. We have carefully addressed your comments in this revised version. Please find our 'response to reviewers' document as attached.

---

## [Editor Report · Decision Letter 2]

9 Aug 2022

Experiences of redeployed healthcare workers in the fight against COVID-19 in China: a qualitative study

PONE-D-21-40460R2

Dear Dr. Guo,

We’re pleased to inform you that your manuscript has been judged scientifically suitable for publication and will be formally accepted for publication once it meets all outstanding technical requirements.

Kind regards,

Xin Liu, Ph.D.

Academic Editor

PLOS ONE
---

## [Editor Report · Acceptance letter]

16 Aug 2022

PONE-D-21-40460R2 

Experiences of redeployed healthcare workers in the fight against COVID-19 in China: a qualitative study 

Dear Dr. Guo:

I'm pleased to inform you that your manuscript has been deemed suitable for publication in PLOS ONE. Congratulations! Your manuscript is now with our production department. 

Kind regards, 

on behalf of

Dr. Xin Liu 

Academic Editor

PLOS ONE